# Integrated multiomics of pressure overload in the human heart prioritizes targets relevant to heart failure

Brian R. Lindman[1,17] ✉, Andrew S. Perry [1,17], Michelle L. Lance[2,17], Kaushik Amancherla[1,17], Namju Kim[1,17], Quanhu Sheng [3], Phillip Lin[1], Ryan D. Pfeiffer[2], Eric Farber-Eger [1], William F. Fearon[4], Samir Kapadia[5], Dharam J. Kumbhani[6], Linda Gillam[7], Ravinder R. Mallugari [1], Deepak K. Gupta[1], Francis J. Miller Jr[8,9], Anna Vatterott[1], Natalie Jackson[1], Yan Ru Su[9], Kelsey Tomasek [9], Tarek Absi[10], Jane E. Freedman[1], Matthew Nayor[11], Saumya Das [12], Quinn S. Wells[1], Marc R. Dweck[13], Robert E. Gerszten [14,15], Eric R. Gamazon [9,18], Nathan R. Tucker[2,18] ✉, Ravi Shah [1,18] ✉ & Sammy Elmariah [16,18] ✉

Pressure overload initiates a series of alterations in the human heart that predate macroscopic organ-level remodeling and downstream heart failure. We study aortic stenosis through integrated proteomic, tissue transcriptomic, and genetic methods to prioritize targets causal in human heart failure. First, we identify the circulating proteome of cardiac remodeling in aortic stenosis, specifying known and previously-unknown mediators of fibrosis, hypertrophy, and oxidative stress, several associated with interstitial fibrosis in a separate cohort ($N = 145$). These signatures are strongly related to clinical outcomes in aortic stenosis ($N = 802$) and in broader at-risk populations in the UK Biobank ($N = 36,668$). We next map this remodeling proteome to myocardial transcription in patients with and without aortic stenosis through single-nuclear transcriptomics, observing broad differential expression of genes encoding this remodeling proteome, featuring fibrosis pathways and metabolic-inflammatory signaling. Finally, integrating our circulating and tissue-specific results with modern genetic approaches, we implicate several targets as causal in heart failure.

Chronic pressure overload initiates a series of macroscopic (hypertrophy), microscopic (fibrosis, apoptosis, innate immune activation, microvascular dysfunction), and molecular alterations in the human heart that precede clinical heart failure (HF)[1-5]. A progression from hypertrophy to HF has been captured in model systems of pressure overload, identifying pathways of hypertrophy and fibrosis[6,7] as targetable mechanisms of HF. Nevertheless, clinical translation has been limited by the accessibility of tissue from the human heart during HF

progression, providing an impetus to find functional, tissue-relevant, non-invasive biomarkers of human HF[8-16]. As paradigmatic conditions, systemic hypertension and aortic stenosis (AS) represent common etiologies of pressure overload that result in HF[17-21]. Specifically, tissue accessibility in patients with AS has enabled studies of myocardial physiology during hypertrophy induction[22,23], suggesting alterations preceding HF may carry diagnostic or prognostic information (e.g., fibrosis). Nevertheless, these tissues are collected during valve

replacement, when the remodeling process has taken hold (potentially with irreversibility). Therefore, development of actionable circulating biomarkers that identify HF risk and reflect human myocardial tissue phenotypes and mechanisms may provide unique, prioritized features for clinical stratification and pathway discovery in human cardiac hypertrophy.

Our goal here was to use a broad proteomic, myocardial transcriptional, and genomic approach to identify mediators causally relevant to cardiac remodeling and HF. Our approach was multi-layered, involving clinical, imaging, tissue-based molecular phenotypes, and human genetics, with a goal of prioritizing physiologic targets relevant in the hypertrophy and HF state (Fig. 1). We quantified a broad circulating proteome in individuals with severe, symptomatic AS awaiting aortic valve replacement (AVR; $N = 809$) to define proteomic signatures of key echocardiographic features of cardiac remodeling (hypertrophy, morphology, and function). To determine the structural and clinical relevance of these findings broadly to HF, we examined the association of these proteins with (1) mortality in individuals with AS post-AVR ($N = 802$), (2) MRI-determined fibrosis ($N = 145$), and (3) incident HF in the UK Biobank ($N = 36,668$). To integrate the proteome with the transcriptome, we next examined whether genes encoding these proteins would be perturbed in AS via single-nuclear RNA-sequencing (snRNA-seq) in 20 individuals (11 with and 9 without AS) to define differential gene expression of key targets within human myocardium at cellular resolution. To provide additional causal validation for identified targets, we employed modern genomic methods (transcriptome- and proteome-wide association) to implicate targets both shared and unique across snRNA-seq and proteomic association studies. This scheme extends recent approaches in multi-organ human aging[24], now using myocardial and circulating samples from the disease process (AS) as well as human genetic approaches to prioritize targets.

## Results

### Cohort characteristics

The AS Biomarker Study ($N = 825$) was split into derivation ($N = 519$) and validation ($N = 306$) samples (baseline characteristics in Table 1). Median age was 83 years, with 44% females and 96% White individuals. Diabetes, coronary disease, and atrial arrhythmias were common, as were abnormalities in cardiac structure (e.g., LV hypertrophy) and systolic (e.g., stroke volume index) and diastolic (e.g., left atrial volume, E/e') function. The single-center CMR AS cohort ($N = 145$) had a median age of 70 years (32% women, all White) across a spectrum of AS severity (mild to severe; Table 2). UK Biobank participants ($N = 36,668$) had a median age of 58 years, 54% were women, and were mostly White (93%; Table 3). On average, the cohort was overweight, with mildly elevated systolic blood pressure and low-density lipoprotein levels, and low prevalence of diabetes. Within the LV myocardial snRNA-seq cohort, the median age of the donor group was 47 years and 55.6% were males. In the AS group, the median was 76 years and 54.5% were males. This is similar to the AS Biomarker Study cohort characteristics (median age of 83 years and comprising 56% males).

### Composite proteomic signatures of myocardial remodeling identify canonical pathways of hypertrophy and fibrosis in aortic stenosis

In the AS Biomarker Study, we identified 3 principal components of 12 parent echocardiographic remodeling traits that accounted for 65% of their overall variation (Fig. 2 and Supplemental Fig. 1). The first principal component ("volumes"; PC1) was weighted on LV volumes (with positive weighting ~ larger volumes). The second component ("systolic function"; PC2) was weighted on measures of LV systolic function (positive weighting ~ better function). The third component ("diastolic function"; PC3) was weighted primarily on indices of diastolic function (positive weighting ~ worse function).

We first estimated age, sex, and race adjusted models for each PC as a function of each protein individually (Supplemental Fig. 2 and Supplementary Data 1). We observed many associations with systolic (PC2) and diastolic function (PC3) phenotypes, specifying pathways of fibrosis (e.g., matrix metalloproteinases, epithelial-mesenchymal transition, TGF-β, immunomodulation, and metabolism). While we were limited in pathway analysis by the limited proteomic coverage in the AS Biomarker Study (979 proteins), targets for specific domains (specifically systolic and diastolic function) did include (1) known pathogenic mediators of cardiac hypertrophy, fibrosis, and aging (e.g., THBS2[11,12], FSTL3[25], GDF-15[26]); and (2) proteins epidemiologically linked to adverse outcomes in human HF (e.g., IGFBP7[27,28], IGFBP4[29]). Importantly, this approach also uncovered targets with evidence in model systems, but not widely reported in human aortic stenosis, including pathways of extracellular matrix remodeling and oxidative stress (LTBP2[30,31], COL4A1[32], MATN2[33]), cardiac development and hypertrophy responses (CRIM1[34,35]), and renin-angiotensin metabolism (ACE2[36]).

### Proteomics of cardiac remodeling and human myocardial fibrosis in aortic stenosis

Apart from hypertrophy, an increasing body of literature suggests the importance of myocardial fibrosis in the pathogenesis of AS, potentially a locus for residual risk after valve obstruction is relieved[37–40]. Therefore, we next mapped our "remodeling proteome" to CMR extracellular volume fraction (ECV; a marker of interstitial myocardial fibrosis) in AS. We measured association between 270 proteins that were (1) measured across proteomics platforms between the AS Biomarker Study and Single-center CMR AS cohort and (2) also associated with any of the 3 phenotype PCs at a 5% FDR (results in Supplementary Data 2; 50 proteins significant at a lenient 10% FDR in Fig. 3). We observed many proteomic associations with myocardial ECV and 6-min walk distance, several displaying directional and biological consistency (e.g., higher thrombospondin-2, GDF-15 [MIC-1] ~ higher ECV [more fibrosis] ~ lower walk distance ~ worse diastolic function [PC3 association]) across cohorts. Associated proteins included mediators of myocardial remodeling/fibrosis (FAM3B[41], IGFBP4) and proteins related to fibrosis in other organ systems (but not in aortic stenosis; e.g., layilin and renal epithelial-mesenchymal transition[42], trefoil factor 2, and hepatic fibrosis[43]).

### Proteomic signatures of cardiac remodeling and outcomes across a broad range of HF susceptibility

We next defined proteomic signatures of each of the 3 phenotype PCs using penalized regression. LASSO regression yielded proteomic signatures with reasonable calibration to the parent PC (Pearson $r$ for each signature against parent PC: 0.42–0.56; Supplemental Figs. 3 and 4 and Supplementary Data 3) that recapitulated epidemiologic differences in cardiac morphology and function by sex (Supplemental Fig. 5) with women having smaller LV volumes (Wilcoxon rank-sum $P = 1.1 \times 10^{-14}$), higher LV systolic function (Wilcoxon rank-sum $P = 3.3 \times 10^{-7}$), and reduced LV diastolic function (Wilcoxon rank-sum $P = 0.013$).

In the AS Biomarker study, of 500 participants in the derivation sample, 193 individuals died over a median 3.2 years follow-up (25th–75th percentile 1.3–3.8 years), and of 302 participants in the validation sample, 134 participants died over a median follow-up of 2.7 years (25th–75th percentile 1.1–3.8 years). We estimated survival models for all-cause mortality, both for the phenotype (echocardiography-based) PC score and proteomic scores (Fig. 4A and Supplementary Data 4). Models using the phenotype PC scores were only available in the derivation sample due to missingness of data in the validation sample to create the phenotype PC scores. Across the deviation and validation samples, we observed increased hazards for

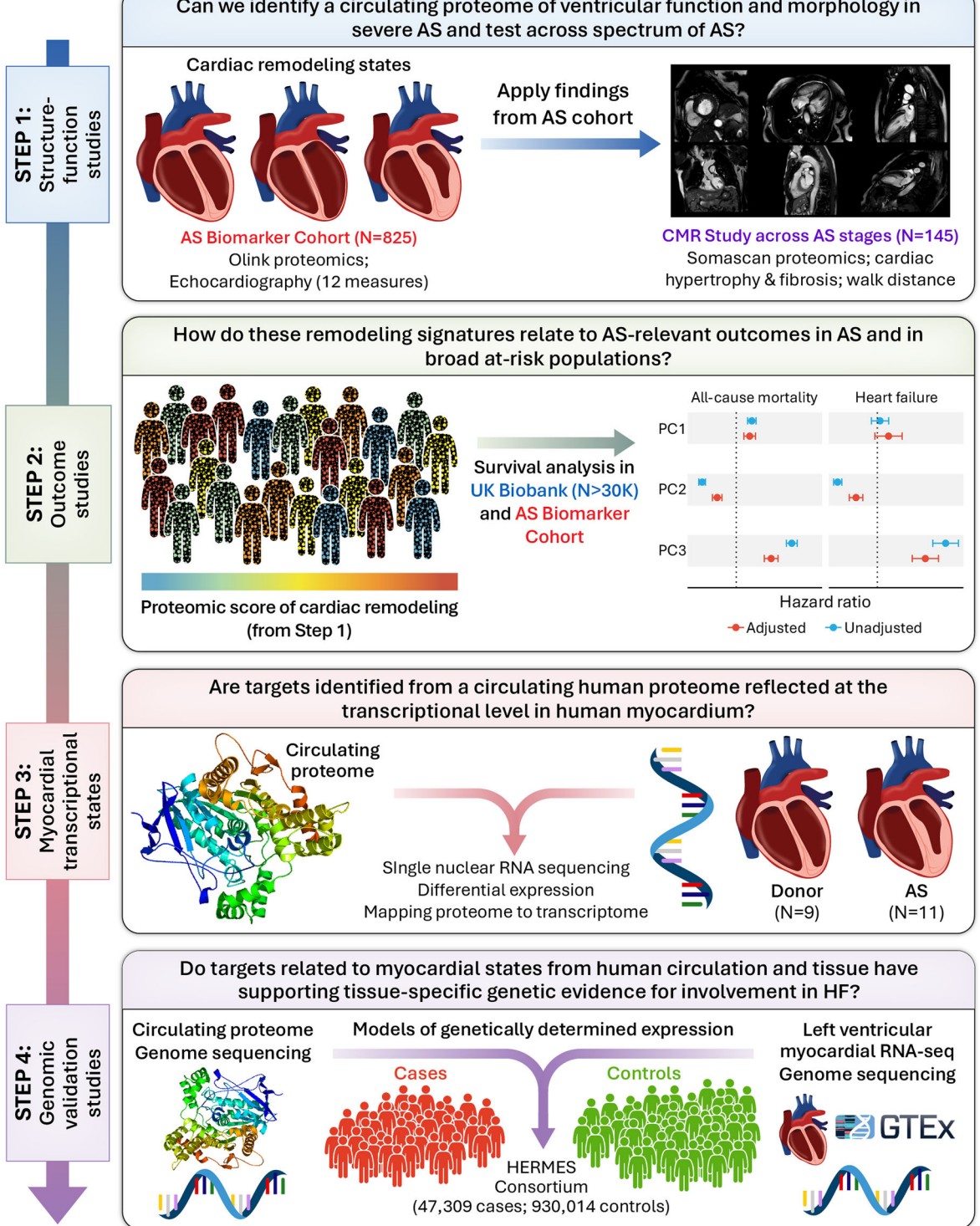

**Fig. 1 | Integration of the circulating proteome and tissue sequencing to prioritize targets of cardiac remodeling.** Overview of our study design. AS aortic stenosis, CMR cardiac magnetic resonance imaging. PC1 = LV Volumes; PC2 = LV Systolic function; PC3 = LV Diastolic function.

components representing a more dilated LV morphology (PC1) and more advanced diastolic dysfunction (PC3), and decreased hazards for the systolic function component (PC2). After multivariable adjustment, our diastolic function proteomic measure (PC3) remained significant ($P < 0.05$) across the samples.

We next calculated proteomic scores in 36,668 individuals in the UK Biobank to evaluate the associations between each proteomic measure and all-cause mortality and incident HF (Table 3). Over a median 13.7-year follow-up (25th–75th percentile 13.0–14.5 years; 3950 mortality events), we observed survival estimates for each proteomic

**Table 1 | Baseline characteristics of AS Biomarker Study (discovery cohort)**

| Characteristic | Overall $N = 825$ | Derivation $N = 519$ | Validation $N = 306$ | P value |
|---|---|---|---|---|
| Age (years) | 83 (77, 88); 0% | 84 (78, 88); 0% | 82 (76, 87); 0% | 0.010 |
| Women | 359 (44%); 0.1% | 233 (45%); 0.2% | 126 (41%); 0% | 0.3 |
| Race | | | | 0.043 |
| Black | 19 (2.3%); 0% | 17 (3.3%); 0% | 2 (0.7%); 0% | |
| Asian | 8 (1.0%); 0% | 6 (1.2%); 0% | 2 (0.7%); 0% | |
| Other | 2 (0.2%); 0% | 1 (0.2%); 0% | 1 (0.3%); 0% | |
| White | 796 (96%); 0% | 495 (95%); 0% | 301 (98%); 0% | |
| Body mass index (kg/m$^2$) | 27.6 (24.3, 31.9); 0% | 27.3 (23.8, 31.4); 0% | 28.0 (24.8, 33.2); 0% | 0.010 |
| Diabetes | 320 (39%); 0.1% | 184 (36%); 0.2% | 136 (44%); 0% | 0.011 |
| Coronary artery disease | 577 (70%); 0.1% | 368 (71%); 0.2% | 209 (68%); 0% | 0.4 |
| History of atrial fibrillation or flutter | 330 (40%); 0.5% | 200 (39%); 0.4% | 130 (43%); 0.7% | 0.2 |
| LV ejection fraction (%) | 61 (53, 66); 3.2% | 61 (53, 66); 0% | 60 (52, 66); 8.5% | 0.3 |
| LV mass index (g/m$^2$) | 107 (90, 126); 18% | 108 (91, 126); 0% | 105 (90, 127); 47% | 0.8 |
| LVH (ASE Grading) | | | | 0.9 |
| Normal | 346 (51%); 18% | 266 (51%); 0% | 80 (50%); 47% | |
| Mild | 118 (17%); 18% | 87 (17%); 0% | 31 (19%); 47% | |
| Moderate | 91 (13%); 18% | 69 (13%); 0% | 22 (14%); 47% | |
| Severe | 125 (18%); 18% | 97 (19%); 0% | 28 (17%); 47% | |
| LV end-systolic internal diameter index (mm/m$^2$) | 15.5 (13.2, 18.7); 18% | 15.5 (13.4, 18.6); 0% | 15.1 (12.8, 18.7); 48% | 0.4 |
| LV end-diastolic internal diameter index (mm/m$^2$) | 23.3 (20.9, 26.1); 18% | 23.5 (21.2, 26.2); 0% | 22.4 (20.2, 25.5); 47% | 0.037 |
| Relative wall thickness | 0.54 (0.45, 0.64); 18% | 0.54 (0.45, 0.64); 0% | 0.55 (0.45, 0.65); 47% | 0.4 |
| LV end-systolic volume index (ml/m$^2$) | 17 (12, 25); 22% | 17 (12, 25); 0% | 18 (12, 27); 59% | 0.5 |
| LV end-diastolic volume index (ml/m$^2$) | 44 (35, 55); 22% | 44 (35, 54); 0% | 45 (35, 55); 59% | 0.8 |
| LV tissue Doppler S lateral annulus (m/s) | 0.065 (0.053, 0.078); 22% | 0.064 (0.053, 0.078); 0% | 0.065 (0.054, 0.076); 59% | 0.5 |
| Stroke volume index (ml/m$^2$) | 37 (29, 44); 16% | 37 (30, 45); 0% | 35 (28, 42); 42% | 0.044 |
| Average transmitral E/e' | 18 (14, 24); 26% | 18 (14, 24); 0% | 17 (12, 24); 69% | 0.4 |
| Left atrial volume index (ml/m$^2$) | 36 (27, 48); 17% | 35 (27, 47); 0% | 37 (30, 48); 46% | 0.2 |
| Aortic valve mean gradient (mmHg) | 39 (31, 49); 2.8% | 39 (32, 50); 0% | 38 (30, 49); 7.5% | 0.3 |
| High-sensitivity cardiac troponin (ng/ml) | 25 (16, 41); 2.9% | 25 (16, 41); 3.5% | 25 (16, 41); 2.0% | 0.8 |
| NT-proBNP (pg/ml) | 1,277 (602, 3,281); 2.9% | 1,303 (622, 3,374); 3.5% | 1,238 (557, 3,049); 2.0% | 0.5 |
| eGFR (ml/min/1.73m$^2$) | 58 (45, 75); 0.7% | 58 (45, 75); 0.6% | 58 (44, 75); 1.0% | 0.5 |

Continuous variables are reported as median (25th, 75th percentile); % missing, categorical variables are reported as $n$ (%); % missing. P-values for continuous variables are from Wilcoxon rank-sum tests, $p$-values for categorial variables are from Chi square tests or Fisher's exact test. *eGFR* estimated glomerular filtration rate, *LV* left ventricular, *NT-proBNP* N-terminal pro hormone B-type natriuretic peptide.

score directionally consistent with our AS cohort (e.g., higher diastolic function proteomic score [PC3] associated with greater all-cause mortality and incident HF; Fig. 4B and Supplementary Data 5).

### A global transcriptional architecture of human aortic stenosis identifies overlaps with the circulating proteome

A primary goal of our work was to test the hypothesis that proteome-phenotype associations from a clinical setting would map to molecular states in the AS heart. We performed snRNA-seq of LV myocardial biopsy samples obtained from 11 patients with AS at the time of surgical aortic valve replacement (SAVR), comparing global and cell-specific transcriptomes with those from unused donors for heart transplantation ($N = 9$). A total of 114,288 nuclei were isolated and clustered into 7 broad cell types (Fig. 5A, B). Detailed demographic data of participants is provided in Supplementary Data 6. Compared with control hearts, AS hearts showed a significant increase in endothelial cells (39% in AS vs 23% in control hearts; Fig. 5C), with no significant changes noted in other broad cell types.

We then tested the relation between circulating protein associations with differences in myocardial expression to prioritize downstream targets and pathways for future investigation. To generate this "proteo-transcriptional" map of human aortic stenosis uniting the

data, we focused on target prioritization by delineating which proteins associated with remodeling are markers for given cell types in the heart, hypothesizing that circulating proteins may derive from select cell types for focused validation studies. We selected markers for all major cell types according to AUC > 0.7, log fold change > 0.6, and an FDR < 0.05 from a Wilcoxon rank-sum test, and cross referenced with the 609 proteins associated with remodeling (Supplemental Fig. 6 and Supplementary Data 7). We observed the largest total overlap within macrophages (113 genes), although we recognize that determining whether these targets derive from the heart or circulating monocytes is not possible in the given dataset. Nevertheless, the large number of proteins with cardiomyocyte-enriched expression (52) are more likely to derive from the heart.

We then tested for differential expression across conditions within each cell type and at the pseudobulk level (Supplementary Data 8). Fibroblasts and macrophages had high overlap of differentially expressed genes (FDR < 0.05) with proteins associated with remodeling (Fig. 5D). Genes with higher expression in AS included convergent pathways of fibrosis—many studied in pro-fibrotic mechanisms in other settings—including TGF-beta and extracellular matrix metabolism and fibroblast proliferation (*WNT9A (fibroblast)*[44], *ITGA6 (fibroblast, endothelial)*[45], *AGRN (fibroblast, endothelial)*[46], *CRIM1*

**Table 2 | Baseline characteristics of the single-center CMR aortic stenosis cohort**

| Characteristic | N | Value |
|---|---|---|
| Age (years) | 145 | 70 (65, 76) |
| Women | 145 | 46 (32%) |
| Aortic stenosis severity | 145 | |
| Mild | | 27 (19%) |
| Moderate | | 39 (27%) |
| Severe | | 46 (32%) |
| Severe/AVR | | 33 (23%) |
| Aortic valve peak velocity (m/s) | 145 | 2.90 (3.33, 4.37) |
| Mean aortic valve gradient (mmHg) | 145 | 35 (23, 44) |
| LV mass index (g/m²) | 145 | 88 (73, 100) |
| Native T1 time (ms) | 140 | 1180 (1,159, 1,210) |
| LV ejection fraction (%) | 145 | 67 (63, 71) |
| Extracellular volume fraction (%) | 140 | 27.81 (25.95, 29.70) |
| Indexed fibrosis volume (ml/m²) | 140 | 23 (19, 27) |
| Mid-wall late gadolinium enhancement | 145 | 41 (28%) |
| Late gadolinium enhancement (%) | 145 | 0.0 (0.0, 1.7) |
| Mitral valve E wave deceleration time (ms) | 143 | 203 (169, 245) |
| E/A mean ratio | 141 | 0.87 (0.72, 1.16) |
| E/e' ratio | 144 | 12.8 (10.7, 17.0) |
| Left atrial volume index (ml/m²) | 133 | 34 (25, 43) |
| Left ventricular longitudinal function (mm) | 144 | 12.02 (10.11, 13.97) |
| 6-min walk distance (meters) | 136 | 400 (338, 450) |

Continuous variables are reported as median (25th, 75th percentile), categorical variables are reported as *n* (%).

**Table 3 | Baseline characteristics of the UK Biobank Cohort**

| Characteristic | N = 36,668 |
|---|---|
| Age (years) | 58 (50, 64) |
| Women | 19,792 (54%) |
| Race | |
| Asian | 761 (2.1%) |
| Black | 804 (2.2%) |
| Mixed | 248 (0.7%) |
| Unknown-other | 594 (1.6%) |
| White | 34,261 (93%) |
| Body mass index (kg/m²) | 26.8 (24.2, 29.9) |
| Low-density lipoprotein (mmol/L) | 3.49 (2.91, 4.10) |
| Systolic blood pressure (mm Hg) | 138 (125, 152) |
| Diabetes diagnosed by physician | |
| Do not know | 86 (0.2%) |
| No | 34,439 (94%) |
| Prefer not to answer | 29 (< 0.1%) |
| Yes | 2077 (5.7%) |
| Townsend Deprivation Index | −2.1 (−3.6, 0.8) |
| Smoking | |
| Current | 3861 (11%) |
| Never \| No Answer | 20,081 (55%) |
| Previous | 12,689 (35%) |
| Alcohol use | |
| Current | 33,417 (91%) |
| Never \| No Answer | 1754 (4.8%) |
| Previous | 1460 (4.0%) |

Continuous variables are reported as median (25th, 75th percentile), categorical variables are reported as *n* (%). The number of missing observations for each variables is as follows: age = 0; women = 0; body mass index = 187; low-density lipoprotein = 1887; systolic blood pressure = 2237; diabetes diagnosed by physician = 37; Townsend Deprivation Index = 51; smoking = 37; alcohol use = 37.

(cardiomyocyte, fibroblast)[47], promotion of epithelial-mesenchymal transition (*SEMA4C (endothelial, fibroblast)*[48], *LAYN (endothelial, fibroblast)*[42], pro-inflammatory phospholipid signaling (*ENPP2/ATX*[49]), and TNF-mediated inflammation, among others. On the other hand, genes downregulated in AS include (1) mediators protective in fibrosis, hypertrophy, or related inflammatory mechanisms (e.g., *EREG*[50], *PTX3 (fibroblast)*[51–53], *HMOX1 (macrophage)*[54], *OPN/SPP1 (neuronal)*[55], *VSIG4 (cardiomyocyte, macrophage)*[56] or (2) those whose absence may benefit remodeling (e.g., *IL1R1 (multiple cell types)*, where fibroblast-specific deletion limits post-infarct remodeling[57]; *NPPC (fibroblast)*, whose deletion reduces fibrosis in diabetic cardiomyopathy[58]). Cell-specific differential expression—when taken with clinical associations—appeared unifying for some mediators: for example, layilin (*LAYN*)—related to adverse echocardiographic remodeling, CMR fibrosis, and differentially expressed at the transcriptional level—exhibited upregulation in both fibroblast and endothelial populations, consistent with potential function in epithelial-mesenchymal transitions[42]. Furthermore, we observed fibroblast-specific downregulation of *PANK1* in AS, which encodes a key enzyme in the coenzyme A pathway, whose deletion demonstrated adverse cardiac remodeling in pressure overload in preclinical models[59] and has recently been mechanistically-linked to the cardiac-specific pharmacologic effects of SGLT2 inihbitors[60]. Of note, *NPPB* expression was not significantly increased in AS myocardium, consistent with a lack of strong correlation between cardiomyocyte expression of *NPPB* and plasma NT-proBNP levels in AS (Supplemental Fig. 7).

We then performed gene ontology analysis to identify convergent functions of gene sets identified from differentially expressed genes in both pseudobulk and a cell-specific manner. Terms include those associated with extracellular matrix remodeling, immune responses, modulation of oxidative stress, and central metabolic pathways (Cori cycle, fatty acids, and oxidative phosphorylation). These global responses were accompanied by differences across cell types, many of which are centered on cell-specific modulation of metabolic, angiogenic, and inflammatory pathways (Fig. 5E and Supplementary Data 9).

**Transcriptome-wide and proteome-wide association studies genetically implicate targets identified through circulating and tissue-based studies in human HF**

A key hypothesis of our multi-layered approach was that an approach integrating circulating markers of susceptibility (proteome) and the organ-specific alterations (single nuclear transcriptome) would provide causal targets relevant to human HF. We constructed polygenic instruments of the circulating proteome as well as left ventricular-specific transcriptional profiles (based on 386 left ventricular samples in the GTEx Portal). We used these instruments in transcriptome- and proteome-wide association studies (TWAS, PWAS, respectively; see "**Methods**") to identify targets prioritized by our proteo-transcriptional approaches causal for HF in a large recent genome-wide association study (HERMES Consortium; 47,309 cases, 930,014 controls[61]; Supplementary Data 10 and 11). We tested 310 targets associated with at least one of the three remodeling PCs at 5% FDR and with differential expression (pseudobulk or cell-specific) in the snRNA-seq analysis. We observed strong genetic causal evidence implicating targets identified by our multi-layered approach in HF (Fig. 5F, G). Importantly, several top enriched targets had previous evidence in support of HF-related mechanisms, including hypertrophy and fibrosis (*SIRPA*[62], *CDON*[63]), autophagy (*DBI*[64]), and metabolism (*ANGTPL4*[65]), among others.

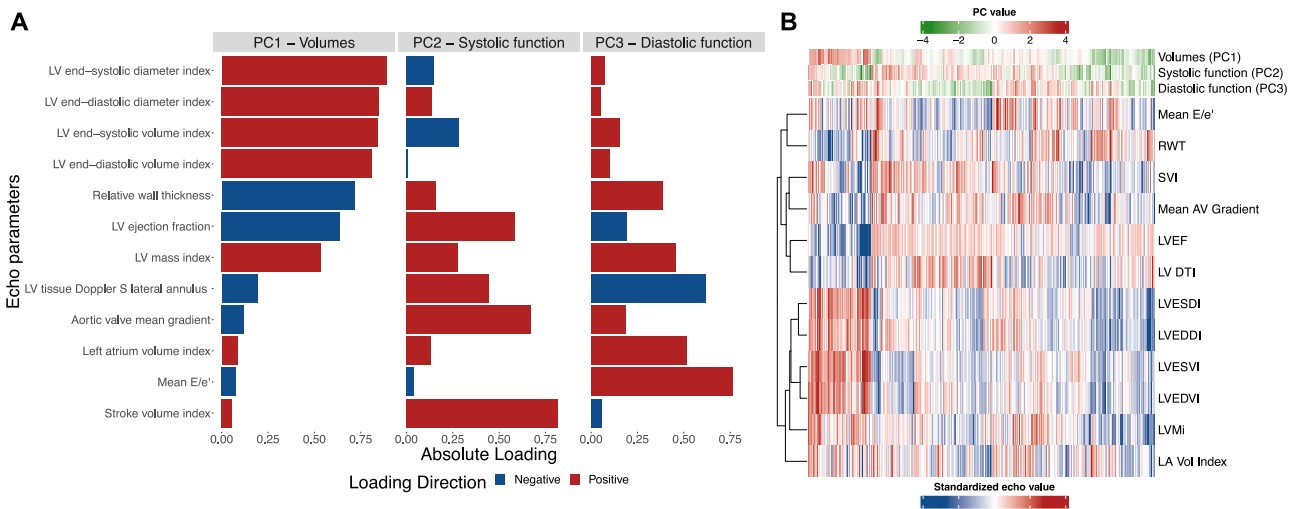

**Fig. 2 | Principal component analysis generates 3 phenotypic components of cardiac remodeling.** We used principal component analysis (PCA) on 12 echocardiographic measures of cardiac structure and function in the discovery cohort derivation sample ($N = 519$) and identified 3 composite axes of remodeling that accounted for 65% of the variance. **A** Loadings for each of the 3 principal components. **B** Heatmap demonstrating individual level data on the 12 echocardiographic measures and participants' corresponding principal component value. Source data are provided as a Source Data file. PC principal component, LV left ventricular, RWT relative wall thickness, SVI stroke volume index, AV aortic valve, LVEF left ventricular ejection fraction, LV DTI LV tissue Doppler S lateral annulus, LVESDI left ventricular end-systolic diameter index, LVEDDI left ventricular end-diastolic diameter index, LVESVI left ventricular end-systolic volume index, LVEDVI left ventricular end-diastolic volume index, LVMi left ventricular mass index, LA Vol left atrial volume.

When we examined left ventricular-specific transcript models, we observed concordance between genetic and snRNA-seq hits across physiologically plausible systems, such as fibrosis and hypertrophy mediators (*CRIM1*[35], *PDCD5*[66]) and mitochondrial metabolism (*GRPEL1*[67]). Finally, recognizing differences in coverage across the proteome and transcriptome, we found three targets common to both PWAS and TWAS with supportive evidence in cardiovascular disease, specifically *MIF* (inflammation[68]), *HEXIM1* (cardiac development[69]), and *ANXA4* (electrophysiologic properties[70]). Importantly, to our knowledge, these (and others not discussed here but implicated by proteo-transcriptional targets and PWAS/TWAS studies; Supplementary Data 10 and 11) represent previously unrecognized avenues for therapeutic modulation in HF.

## Discussion

Despite the success of guideline-directed medical therapy in HF, high residual clinical risk has driven a search for additional targetable mechanisms of myocardial remodeling. Seminal results from model systems (e.g., beta-adrenergic system, angiotensin II signaling, etc.) have been recently complemented by an emergence of human "omics" studies, aimed at capturing complex heterogeneity in human HF. The inaccessibility of myocardial tissue sampling during HF development necessitates use of other more accessible markers for patient-based discovery (e.g., circulating plasma). While genomics has yielded high value targets for Mendelian susceptibility for dilated cardiomyopathy, genetic effects on more common acquired variations of HF (e.g., valvular or hypertensive) are more limited. In addition, mechanisms that address HF and cardiac remodeling (e.g., GLP-1 receptor agonist[71], SGLT2 inhibition[72]) may target non-cardiac mechanisms. In this context, identifying functional biomarkers of acquired HF from clinically relevant populations across the spectrum of HF susceptibility and mapping onto the human heart through transcriptional and genetic approaches may offer a way of prioritizing potential cardiac-specific targets in HF development and progression for study.

Here, we address this call in human AS through four integrated steps: (1) defining key phenotypic components of myocardial remodeling and their proteomic signatures in severe AS using multi-modality imaging and circulating proteomics; (2) estimating HF risk based on this remodeling proteome in settings post-AVR and more broadly in a developmental stage of HF (UK Biobank); (3) mapping genes encoding these proteins into the myocardium from patients with and without AS at single nuclear resolution to assess proteo-transcriptional concordance; and (4) using modern genomic methods leveraging human left ventricular myocardial expression and proteome-wide association to provide causal validation. This approach comes in the context of numerous proteomic studies to identify and characterize the risk of HF and the mechanisms of left ventricular remodeling[11–16]. Across 3 major domains of myocardial remodeling (morphology, systolic and diastolic function), we identified proteomic patterns that displayed biological relevance and plausibility and concordance with in vivo CMR phenotypes (e.g., interstitial fibrosis) and prior annotation in HF (THBS2[11,12], FSTL3[25], GDF-15[26], IGFBP7[27,28], IGFBP4[29]). Similar to prior investigations, we did not observe any statistically significant relationships between plasma proteins and LGE (replacement fibrosis[73]). Moreover, the proteomic patterns linked to cardiac structure-function also identified near-term outcomes post-AVR and longer-term outcomes in a lower risk population (UK Biobank), underscoring the importance of proteomic profiles across a HF risk spectrum. Collectively, these results expand extant "omic" studies of AS in humans (limited primarily to valve tissue[74–76], with small sample sizes[77] or limited proteomic coverage[78]) and suggest potential for an approach uniting clinical, phenotypic, and transcriptional states for pathway discovery in HF.

Of key importance in our work was the push to integrate across the proteome, transcriptome, and tissue-relevant genome to prioritize targets of causal (and therefore likely functional) relevance. Until recently, the idea that the circulating proteome and tissue transcriptome could be used in joint inference has been considered difficult[24,79]. In the cardiovascular space, Chan and colleagues recently mapped a circulating proteome of post-myocardial infarction clinical events to differentially expressed genes from single-cell/single-nuclear RNA-seq from mice and humans with HF[79], following a similar workflow and identifying several key canonical markers consistent with our study (e.g., *THBS2*, *FSTL3*). Nevertheless, these approaches remain

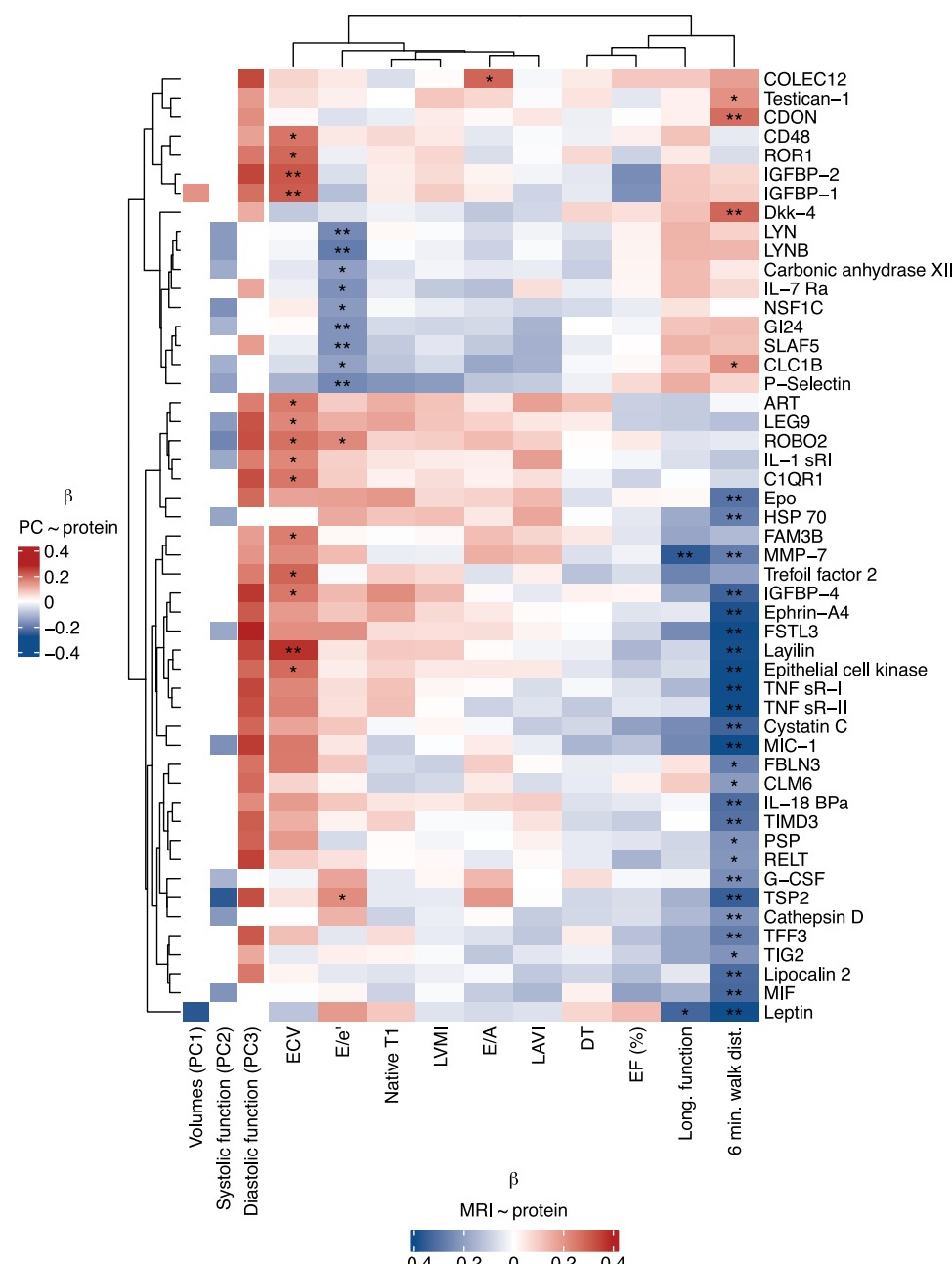

**Fig. 3 | Validation of proteomic markers of remodeling in a CMR-based cohort.** We examined the relations of the 270 proteins related to any echocardiographic component of remodeling (Supplemental Fig. 2) with cardiac magnetic resonance imaging (CMR) based measures of remodeling and fibrosis. For visualization, proteins related to any CMR measure with an FDR < 0.1 (Benjamini−Hochberg) are presented. Source data are provided as a Source Data file. ECV extracellular volume, LVMI left ventricular mass index, LAVI left atrial volume index, DT deceleration time, EF left ventricular ejection fraction. * FDR < 0.1; ** FDR < 0.05.

scant in the HF literature. Here, we observed broad and cell-specific differential expression patterns across canonical pathways of remodeling, namely extracellular matrix remodeling, immune responses, oxidative stress, and central metabolic pathways. Acknowledging mRNA-protein discordance[80], a unique union of data allowed us to test whether proteins related to adverse remodeling phenotypes overlap with differentially expressed targets in the myocardium. This proteo-transcriptional overlap identified a panoply of targets central to remodeling mechanisms (with several genes not widely described in human HF), including fibrosis (*WNT9A*[81], *ITGA6*[45], *AGRN*[46], *CRIM1*[47], *SEMA4C*[48], *LAYN*[42]) and inflammation (*ENPP2/ATX*[49], TNF family members). Importantly, those genes positively associated with poorer global myocardial structure but downregulated in AS myocardium

included genes thought to be protective in fibrosis, hypertrophy, or inflammation (e.g., *EREG*[50], *PTX3*[51–53], *HMOX1*[54], *OPN/SPP1*[55], *VSIG4*[56]) or those whose absence may benefit remodeling during pressure overload (e.g., *IL1R1*, *NPPC*, etc.). Additionally, prioritization of genes based on the integration of the circulating proteome and tissue transcriptome highlights fibroblasts, endothelial cells, and cardiomyocytes potentially as highly relevant drivers of AS pathophysiology.

While these findings extend recent seminal reports linking specific mediators of inflammation and metabolism to fibrosis[23,82] by this "proteo-transcriptional" mapping, examining the circulating or transcriptional states during or even upstream of a disease process has some limitation. Given the role for human genetics in identifying intervenable targets in disease[83], we used modern transcriptome- and

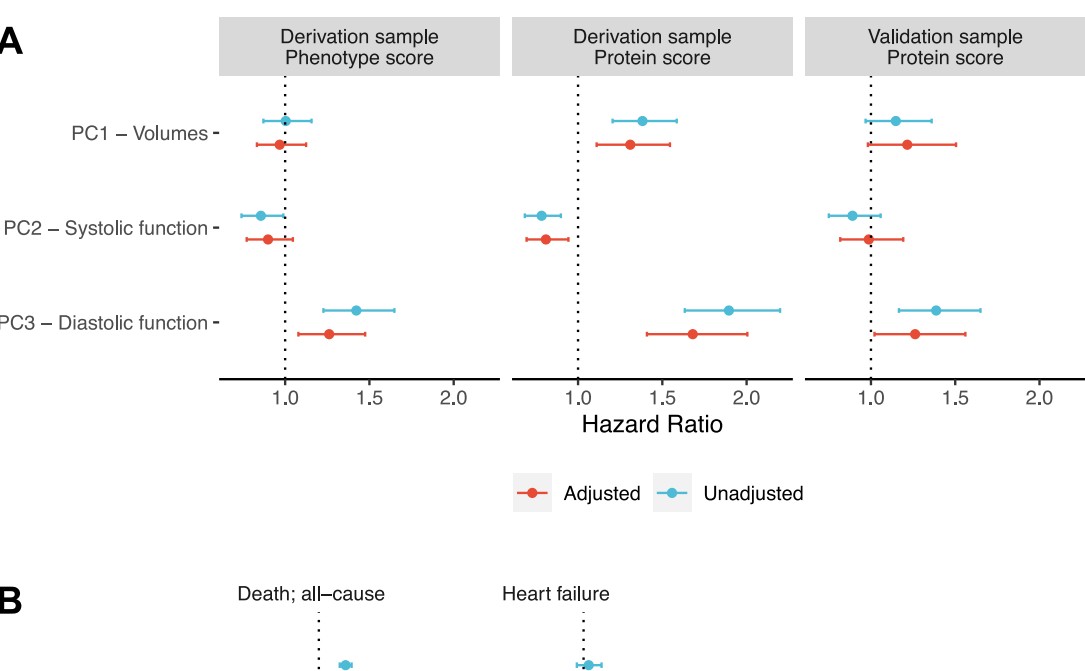

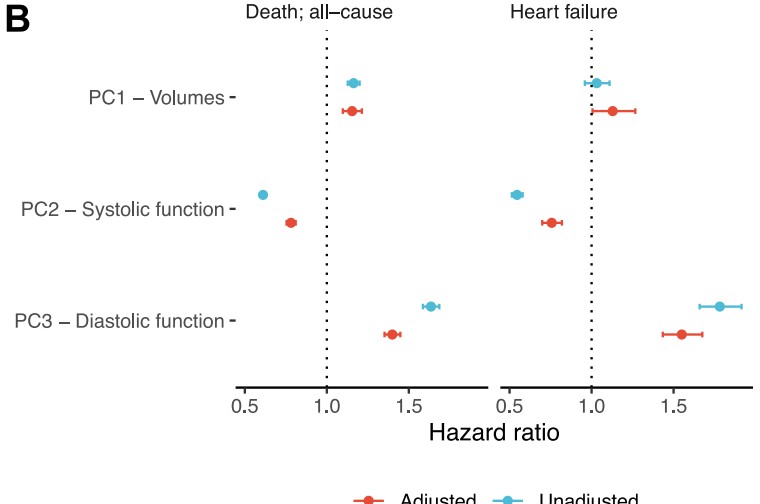

**Fig. 4 | Proteomic signatures of remodeling are related to clinical outcomes in both an AS population and a general "at-risk" population. A** Forest plot of Cox regression results from the discovery cohort relating the proteomic scores to post-TAVI all-cause mortality. Derivation sample *N* = 500 in unadjusted models. Validation sample *N* = 302 in unadjusted models. **B** Forest plot of Cox regression results from the UK Biobank (*N* = 36,668 in unadjusted models) relating proteomic scores of remodeling to all-cause mortality and incident heart failure. Source data are provided as a Source Data file.

proteome-wide approaches (TWAS/PWAS; see "**Methods**") to provide causal evidence to targets identified via proteomic or snRNA-seq studies. These studies supported the multi-layered approach here, demonstrating a strong enrichment for proteo-transcriptional targets in large human HF genome-wide association studies. Identified targets (including three that were implicated by PWAS, TWAS, and our clinical studies) generally had model system evidence supporting involvement in HF or HF-relevant mechanisms, but to our knowledge have not been sources of intervention currently in AS or HF. Furthermore, these studies allowed us to examine causal relevance of both circulating and tissue-specific (e.g., LV) targets, accessible by leveraging genetic prediction of protein levels within circulation or of LV transcript expression (within GTEx). While this approach is not exhaustive of all biologically relevant targets and protein secretion/function outside the heart may be critical to disease (e.g., SGLT2 inhibition, GLP-1 receptor agonist, etc.), the identification of biologically plausible targets with human genetic support offers validity to the multi-parametric approach we undertook.

Our results highlight innovations in multi-level integration between the phenome, proteome, transcriptome, and the genome to tier prioritization of plausible targets for intervention. Nevertheless, our results should be contextualized in view of key limitations. While our data unites AS patient samples across tissue and plasma as well as population-based studies (with important differences in age across cohorts), replication of proteomic findings, human genetic support, and biological plausibility bolster validity. Larger studies across a range of comorbidities, harmonized and innovative proteomics platforms, tissue and imaging analyses, and with pre- and post-assessment after valve replacement are needed. While we attempted to adjust for multiple confounders in regression modeling, residual confounding remains possible. Cardiac amyloidosis co-exists in older adults with severe AS in approximately 10–15% of cases and we are unable to parse out the effect of amyloid deposition. Differential causes of elevated ECV (e.g., amyloidosis) were not explored with endomyocardial biopsy in our study, though our range of ECV was lower than classically reported in this condition. Additional effects of concomitant medications directed toward HF management may also impact proteomic patterns; further studies adjusted for medications in AS are warranted. Nevertheless, the relation of the signature to long-term HF development supports its relevance to HF. While we did not observe

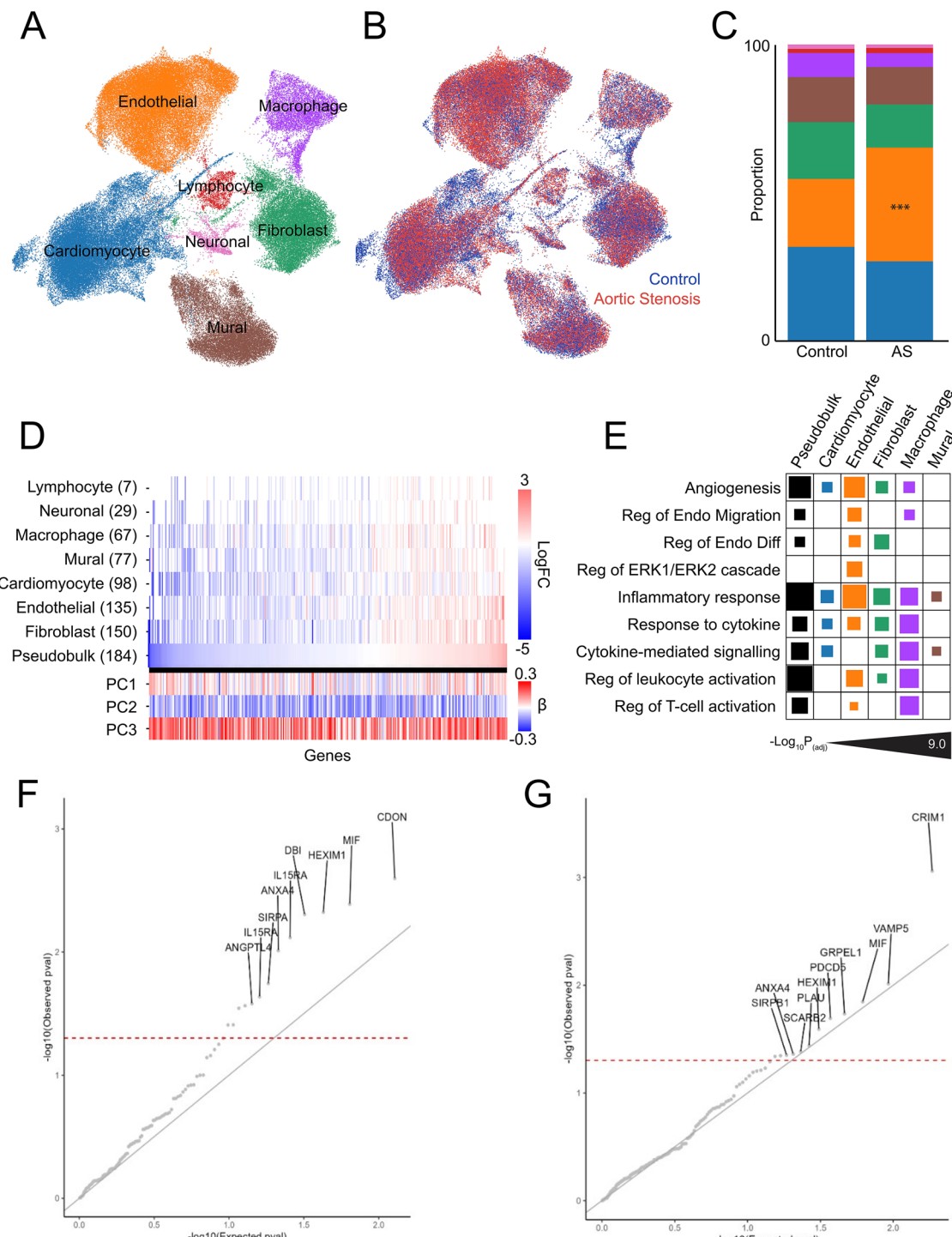

**Fig. 5 | Proteo-transcriptional architecture of human aortic stenosis.** UMAP of cell types (**A**) and condition (**B**) defined by single-nuclear RNA-sequencing (114,288 nuclei across 7 broad clusters). **C** Cell composition by condition identifies a significant increase in endothelial cells in AS hearts; *** represents significance. **D** Heatmap representing the differentially expressed genes associated with remodeling across pseudobulk and cell-specific analyses. Numbers adjacent to the cell type label represent the total number of significant genes, inclusive of those which overlap across comparisons. **E** Select terms from Gene Ontology analysis of differentially expressed genes associated with remodeling, which highlight cell type-specific pro-angiogenic and pro-inflammatory processes. Full tabular results for all analyses are available in Supplementary Data 9. **F** Quantile-quantile plot showing

the association of genetically determined circulating protein expression with HF. A leftward shift from the gray line shows departure from the null (standard uniform) distribution. The red dashed line denotes $P < 0.05$. The protein IL15RA is shown twice as it is tagged by 2 SOMAmers, each of which shows an association with HF. **G** Quantile-quantile plot showing the association of genetically determined gene expression in the left ventricle with HF. A leftward shift from the gray line shows departure from the null (standard uniform) distribution. The red dashed line denotes $P < 0.05$. Note *MIF*, *HEXIM1*, and *ANXA4* also show $P < 0.05$ in the association between the genetic component of the respective *protein* expression and HF. Source data are provided as a Source Data file.

differential gene expression of natriuretic peptides (*NPPB*) in AS versus control, consistent with the absence of strong correlation between circulating NT-proBNP levels and cardiomyocyte *NPPB* expression in AS patients (Supplemental Fig. 7 and Supplementary Data 6). Physiologic changes prior to tissue sampling (e.g., brain death in donors, hemodynamic optimization prior to surgical AVR in AS patients) as well as lack of end-stage myocardial disease in the AS cohort may account for heterogeneity. We did not exclude participants with AS in the UK Biobank, although we expect AS to be of low prevalence. Use of endomyocardial biopsies post-TAVI may be useful to resolve these differences (performed in small studies[84]). Additionally, proteomic studies at the bulk or single cell level in human myocardium to compare individuals with and without AS would be further revealing beyond the proteo-transcriptional approach presented here. Given potential horizontal pleiotropic and confounding effects, definitive conclusions on causality beyond TWAS approaches here will require additional supportive work beyond our TWAS approach (e.g., functional studies). We envision that these approaches can be used to select genes relevant for gain- and loss-of-function approaches in model studies or in vitro organoids for further characterization.

In conclusion, we present a tiered approach to discovery leveraging echocardiography, MRI tissue imaging, large cohort validation, and tissue single nuclear transcription and comprehensive genetic approaches. Our approach not only identified a prognostic signature of human HF in AS, encompassing relevant known and previously unknown pathways of global and tissue-level remodeling reflected in human heart tissue in AS and supported by broad genetic causal support. Given limitations of myocardial sampling in longitudinal clinical encounters, these results offer a paradigm to identify, validate, and map the circulating human remodeling proteome during myocardial pressure overload to window reported associations to potentially causal, tissue-relevant mediators of cardiac remodeling. Future applications of this approach across a broader diversity of HF phenotypes, a broader proteomic space, and tissue availability may offer a prioritizing framework for discovery in HF. We hope that the findings of our study and approaches undertaken here will serve as a resource for the larger scientific community to accelerate discovery in cardiovascular disease.

## Methods
### Ethical approvals
This study complies with all relevant ethical regulations. All subjects provided written informed consent, and the Institutional Review Boards at Vanderbilt University Medical Center and Massachusetts General Hospital approved this study. Written informed consent was obtained to publish indirect identifiers. UK Biobank analyses were approved by Research Proposal 57492. Data from the HERMES Consortium used in genomic analyses are published[85].

### Study populations
**AS biomarker study.** Our discovery cohort was derived from a multicenter observational cohort study of 825 adults with symptomatic, severe aortic stenosis (AS) who were prospectively enrolled prior to transcatheter aortic valve implantation (TAVI) between May 2014 and February 2017 (methods reproduced with some modification from our prior work[86–88]). Severe AS was defined according to current guidelines: peak velocity ≥4 m/s, indexed aortic valve (AV) area <0.6 cm$^2$/m$^2$, or mean gradient ≥40 mmHg[89,90]. All participants underwent TAVI. We divided this cohort into derivation (*N* = 519) and validation (*N* = 306) samples. The derivation sample included only participants with complete data across 12 echocardiographic measures of left ventricular (LV) remodeling (by transthoracic echocardiography prior to TAVI; median 39 days, 25th–75th percentile 21–72 days). Echocardiograms were analyzed at a core laboratory to quantify LV structure and function according to American and European guidelines[91], including the

following 12 measures of LV remodeling: LV ejection fraction (LVEF), LV stroke volume index, LV internal dimensions and volumes at end-diastole and end-systole, LV mass, relative wall thickness, mean transmitral E/e', tissue Doppler S velocity of the lateral mitral annulus, left atrial volume, and mean AV gradient[91]. Of the 825 participants included in the study, 809 participants had a blood sample available for proteomic analysis and 802 had vital status assessed between March and June 2020 and, thus, were included in Cox regressions.

**Single-center cardiac magnetic resonance (CMR) AS cohort.** This cohort included 145 individuals across the spectrum of AS (mild to severe) enrolled at Edinburgh Heart Centre from March 2012 to August 2014 with CMR, echocardiography, 6-min walk test, and proteomics (Somalogic). (CMR and methods as described in our primary work are reproduced here for scientific rigor[1]). Exclusion criteria for the cohort included other significant valvular heart disease (≥ moderate severity), significant comorbidities limiting life expectancy to <12 months, contraindications to gadolinium-enhanced CMR, and acquired or inherited nonischemic cardiomyopathy (as assessed by clinical history or CMR). The severity of AS and indices of diastolic function were assessed using transthoracic echocardiography performed by a single experienced echocardiographer according to American and European guidelines[91,92]. CMR imaging was performed using a 3-T scanner (MAGNETOM Verio, Siemens AG, Erlangen, Germany). Short-axis cine images were acquired and used to calculate ventricular volumes, mass, and function. Left ventricular (LV) mass was indexed to body surface area (Du Bois formula). LV longitudinal function was quantified as mitral annular displacement between end-systole and end-diastole. CMR interstitial fibrosis was assessed using native T1 and extracellular volume (ECV) assessments derived from T1 mapping data acquired before and after gadolinium contrast administration. ECV was used as our principal marker of myocardial fibrosis, given its quantitative nature and powerful prognostic information in patients with AS[93]. Detailed imaging protocols have been previously described[1,94]. The study was conducted in accordance with the Declaration of Helsinki and approved by the local research review committee, and informed consent was obtained from all individuals.

**UK Biobank.** The UK Biobank is a population-based cohort recruited between 2006 and 2010[35]. Our sample began with ≈54,000 UK Biobank participants whose circulating proteins were quantified using the Olink Explore 1536 panel[95,96]. We included 36,668 participants without missing data on the 75 proteins used in our proteomic signatures of LV remodeling, which signatures were developed in the AS Biomarker Study.

### Proteomics
Peripheral, venous blood samples (*N* = 825) were obtained prior to TAVI from participants in the AS Biomarker Study, and quantification of the circulating proteome was performed using the Olink Explore 1536 platform in 3 batches[97]. We included 1 sample for each participant. For quality control purposes, 2 pooled plasma samples (pooled from all 825 study participants) were included across all batches ("bridging" samples). A batch effect was detected and corrected by median normalization using the plasma "bridging" samples and setting the batch with the most samples as the reference batch as described[97]. From the 1472 proteins quantified, 258 proteins were removed due to technical warnings on the Oncology panel, 120 proteins for >25% of values reported below the limit of detection, and 107 proteins with a coefficient of variation >40%, consistent with our prior work[97]. We excluded NT-proBNP, BNP, and troponin to facilitate identification of new biomarkers of remodeling. As TNF, IL6, and CXCL6 were measured across multiple panels in Olink Explore, we randomly selected a single assay for inclusion. Overall, 979 circulating proteins were included for analysis in the AS Biomarker Study. Protein levels were expressed in

Normalized Protein eXpression units, and scaled to mean of 0, and variance 1 across the dataset.

In the single-center CMR AS cohort, the circulating proteome was quantified using an aptamer-based approach (Somalogic panel, $N \approx 1300$ aptamers). Mapping of proteins across Olink and Somalogic platforms was performed using UniProt identifier, and only Somalogic aptamers uniquely associated with one UniProt identifier were used. UK Biobank blood samples were obtained at the initial assessment visit (2006–2010)[96], and proteomics quantification was performed using Olink Explore 1536 (same as in AS Biomarker Study)[96].

**Myocardial tissue sampling and single nuclear RNA sequencing**
Following informed consent, myocardial tissue was obtained with a scalpel from the basal septum of the left ventricle from patients ($N = 11$) undergoing SAVR for AS or unmatched donor hearts from 2011 to 2012 ($N = 9$; hearts not used for transplantation) and immediately snap frozen in liquid nitrogen. The interventricular septum was directly visualized by the operating surgeon (e.g., SAVR) or heart procurement surgeon (e.g., donor hearts) prior to tissue procurement. The tissue was subsequently stored in a −80 °C freezer until used for analysis. All samples were partial septum samples and non-transmural.

Single nuclear suspensions from myocardial tissue were prepared as described previously[61,98]. Briefly, frozen ventricular tissue was mounted onto a CryoStat using Optimal Cutting Temperature (OCT) compound, followed by 100 μm sectioning. OCT was removed, and sections were placed in cold lysis buffer, followed by gentle Dounce homogenization to release intact nuclei. After pelleting at $40 \times g$ for 1 min to remove large debris, the suspension underwent sequential filtering using 100 μm and 20 μm filters. The filtered suspension was centrifuged at $550 \times g$ for 5 min, washed in a nuclear wash buffer, and recentrifuged at $550 \times g$ for 5 min. The nuclear wash buffer was removed via aspiration and the nuclei were resuspended in cold nuclear resuspension buffer containing RNase inhibitors. Nuclei were stained with trypan blue and counted using a hemocytometer.

Nuclei were loaded into a 10× Genomics microfluidic platform (Chromium Next GEM Single Cell Multiome ATAC + Gene Expression), targeting an estimated recovery of 8000 nuclei per sample. Libraries were processed according to the manufacturer's instructions.

**Statistical methods**
**Circulating proteomic architecture of cardiac remodeling in human AS.** The overall flow of our study is shown in Fig. 1. We used principal component analysis (PCA) in the derivation sample of the AS Biomarker Study ($N = 519$) to summarize 12 echocardiographic measures of LV remodeling given collinearities across echocardiographic traits. The rationale for this approach was to group together those domains that are physiologically similar (hypothesizing that they may have shared proteomic association). We retained 3 principal components (PCs) in this analysis based on scree plot. Scores from the 3 PCs were used as dependent variables in cross-validated penalized regression (LASSO; least absolute shrinkage and selection operator) as a function of 979 circulating proteins (proteins were scaled to a mean of 0 and variance of 1 prior to use in models) to generate proteomic signatures for each remodeling PC (*caret* in R[99]; across individuals with proteomic data available, $N = 503$). We also fit linear models for each remodeling PC score as a function of each individual protein, adjusted for age, sex, and race (with 5% false discovery rate [FDR] for multiplicity, Benjamini–Hochberg). We constructed a linear combination of protein-specific regularized beta coefficient from the LASSO regression and the mean-centered, standardized protein levels in the AS Biomarker Study and UKBB to construct proteomic signatures for survival modeling.

**Relation to CMR-defined phenotypes in AS.** We selected proteins associated with any of the 3 remodeling PCs (at an FDR < 5%) that were measured in the Single-center CMR AS cohort (cross-platform matching Olink to Somalogic via UniProt identifier) for association with LV remodeling and fibrosis (by CMR) and functional capacity (6-min walk distance) in the single-center CMR AS cohort. We estimated age- and sex-adjusted linear models using individual proteins as independent variables in separate models for each remodeling, fibrosis, or functional trait (dependent variables).

**Association with clinical outcomes in AS and at-risk HF.** Clinical outcomes were available in both the AS Biomarker Study ($N = 500$ for derivation sample unadjusted models; $N = 302$ for validation sample unadjusted models) and the UK Biobank ($N = 36,668$ for unadjusted models). We estimated Cox models for all-cause mortality in the AS Biomarker Study (only outcome adjudicated), and all-cause mortality and incident HF in UK Biobank. Deaths in UK Biobank were defined by death registry data (UK Biobank Data Field 40000). Incident HF in the UK Biobank was defined by ICD10 codes grouped into "phecodes" using the PheWAS package in R[100], with prevalent cases removed from analyses (prevalent cases defined by self-report or physician diagnoses; Data Fields 20002, 6150). Censoring dates in UK Biobank were determined using each participant's location of initial assessment (UK Biobank Data Field 54) and the region-specific censor dates provided by the UK Biobank. Models for death were censored at the region-specific censor dates. Models for incident HF were censored at the time of death or the region-specific censor date. In the AS Biomarker Study, we adjusted models for age, sex, body mass index (BMI), diabetes (hemoglobin A1c ≥ 6.5% or documented history), coronary artery disease (prior myocardial infarction, prior revascularization, or atherosclerotic disease in ≥1 coronary artery), history of atrial fibrillation/flutter, and estimated glomerular filtration rate (eGFR). In UK Biobank, we adjusted for age, sex (UK Biobank Data Field 31), race, BMI, systolic blood pressure, diabetes, Townsend Deprivation Index[101], smoking, alcohol use, and low-density lipoprotein.

**Single nuclear RNA sequencing processing**
We have reproduced methods with minimal modification from our prior work[61,98,102,103] to ensure scientific rigor and reproducibility, with attribution by this statement. The sequencing files were demultiplexed using Cellranger-ARC *mkfastq* (v2.0.0). The RNA reads were then trimmed using Cutadapt (v2.8) with default parameters to remove homopolymers (A30, T30, G30, and C30) and the template switch oligo (CCCATGTACTCTGCGTTGATACCACTGCTT) and its complement (AAGCAGTGGTATCAACGCAGAGTACATGGG).

To generate an RNA count matrix, Cellranger (v7.1.0) *count* was used with default settings to align the trimmed reads to the human genome (GRCh38) and CellBender (v2.0)[104] was used with default settings to filter samples for ambient RNA byproducts and call valid cell barcodes. In total, 228,367 nuclei were identified across the 20 samples. Further analysis was performed using Scanpy v1.9.3. Data for each samples was inspected individually for evaluation of mapping and complexity metrics (Supplemental Figs. 8 and 9).

To identify nuclei with high levels of spliced reads indicative of increased cytosolic fragment contamination, we used Scrinvex (v13, https://github.com/getzlab/scrinvex) to assign reads to exonic and intronic regions. We then calculated the ratio of exonic reads to the total number of mapped reads within each sample. Nuclei with an exon/intron ratio greater than the 75th percentile + IQR range were removed from downstream analyses (23,127 nuclei). Doublets were detected using Scrublet[105], a threshold of 0.15 was set based on the simulated distribution, and nuclei with a doublet score >0.15 were removed from the analysis (75,409 cells). Finally, nuclei with reads mapped to fewer than 200 genes (6181 nuclei) and nuclei with more than 5% of reads mapped to mitochondrial genes (33,034 nuclei) were removed (Supplemental Figs. 8 and 9).

Counts were normalized to $10^3$ counts per nucleus and log-normalized. Highly variable genes were identified for clustering (minimum mean 0.0125, maximum mean 3, minimum dispersion 0.5, 4283 genes). The effects of the total read count and the percentage of mitochondrial reads were then regressed out using default parameters (Scanpy *regress_out*).

Principal components (PCs) were calculated for highly variable genes and integrated using Harmonypy[106] (version 0.0.9), with default parameters considering each sample as a unique batch. Corrected PCs were used to calculate the nearest neighbors and generate a UMAP representation of the data for two-dimensional visualization purposes. Graph-based clustering was performed using Leiden algorithm at a basal resolution of 0.2.

**Marker gene and cell type identification.** Marker gene and cell type assignments were calculated using Log$_2$-fold change in each cluster versus all others combined, the percentage of cells expressing each gene, and the area under the receiver operator curve (AUC) (SciKit Learn *roc_auc_score*). Genes with an AUC > 0.7 and a log$_2$-fold change >0.6 were considered as potential markers of a given cluster.

**Differential composition and expression analysis.** Differences in cell composition between AS and donor hearts were compared using scCODA[107], a Bayesian model based on the Dirichlet-multinomial model in order to determine statistically credible changes in cell composition. The model formula included condition (AS or Control) as the covariate and neuronal cells were used as the reference control, given highest stability across all samples. Hamilton Monte Carlo sampling was performed with 20,000 iterations. No *P*-values are provided, as this approach is Bayesian in design.

The AnnData file was subsequently converted to a Seurat object using the Seurat disk package. Cell types that could not be classified were excluded from downstream analyses. Differential expression testing between AS and control hearts was performed in two iterations using edgeR glmLRT function: (1) pseudobulk per sample, with summed counts for genes across all nuclei and cell types within a single individual and (2) by cell type, with summed counts for genes across nuclei within each cell type for each individual. *P*-values were adjusted for multiplicity using the Benjamini–Hochberg correction. Genes were considered differentially expressed if they met both an FDR-adjusted *P*-value < 0.05 and an absolute log fold change >1.2. Gene Ontology enrichment analysis was performed with the hypergeometric test (using DAVID) conducted on targets that were both differentially expressed genes within snRNA-seq and significant in the circulating proteome.

**Genetically determined gene expression in heart left ventricle**
We developed in silico genetic models of gene expression using whole-genome DNA and RNA-seq data in human heart left ventricle (N = 386) from the Genotype-Tissue Expression (GTEx) Project (v8)[108]. We utilized normalized gene expression after adjusting for the first 5 genotype-based principal components (representing genomic ancestry), sex, and platform. For model features for a given gene, we considered the SNPs of sufficiently high minor allele frequency (MAF > 0.05) within the *cis*-region (of size determined by cross validation) of the gene. To enhance gene expression modeling relative to single-tissue approaches, we used the JTI approach[109], which leverages the following objective function:

$$\hat{\beta} = \arg\min_{\beta}(1/2) \sum_{i=1}^{n} w_i \left(y_i - x_i^T \beta\right)^2 + \lambda\left(\left(\frac{1-\alpha}{2}\right)||\beta||_2^2 + \alpha||\beta||_1\right).$$

Here, $||\beta||_1$ and $||\beta||_1$ denote the L$_2$ and L$_1$ norm respectively of the effect size $\beta$, $y_i$ is the gene expression of the *i*-th sample, $x_i^T \beta$ is the genetically determined expression, and the weight $w_i$ is derived from a gene-level similarity matrix (defined using the expression measurement from the RNA-seq and regulatory profile from DNase I hypersensitivity data). Hyperparameters were obtained from cross validation. By design, the model training exploits the shared regulation of gene expression across tissues to improve model performance. Notably, the approach shows greater prediction performance in external datasets and higher replication rate than conventional single-tissue approaches[109].

**Association of genetically determined gene expression with HF**
We sought genetic validation for the set of genes significant from the single nucleus RNA-seq differential expression analysis (and significant in the corresponding circulating proteome). We leveraged publicly available data in genomics of HF from the Heart Molecular Epidemiology for Therapeutic Targets (HERMES) Consortium[110]. This dataset is a genome-wide association study (GWAS) meta-analysis across 26 sub-studies involving 47,309 cases and 930,014 controls of European ancestry. We then performed a transcriptome-wide association study (TWAS) of HF using the HERMES SNP-level summary data, testing the association of genetically determined gene expression in left ventricle with HF[111–114].

**Association of genetic component of protein expression with HF**
We also sought genetic validation of the set of proteins significant (FDR < 0.05) in the circulating proteome (and significant in the single cell pseudobulk differential expression analysis). To perform a veritable independent validation, we leveraged a large-scale proteogenomic dataset (N = 7, 213 European ancestry individuals)[115] that does not overlap with the UK Biobank. This dataset consists of protein expression measurements from 4657 SOMAmers mapping to 4435 genes. The genetic models of protein expression had been developed using the same methodology as has been implemented in PrediXcan[114]. We applied the models to GWAS summary data of HF from the HERMES Consortium[110] in a proteome-wide association study (PWAS), allowing us to evaluate the association between the genetic component of protein expression and HF.

**Reporting summary**
Further information on research design is available in the Nature Portfolio Reporting Summary linked to this article.

## Data availability
Olink Proteomics data from the AS Biomarker Cohort is available at https://doi.org/10.6084/m9.figshare.28873238. Because the AS Biomarker and single-center CMR cohorts were collected under prospective protocols that restrict data sharing to de-identified, aggregate results, individual-level records cannot be posted publicly without violating participant-consent agreements and institutional privacy regulations. We have provided aggregate data in our Tables. Investigators may request access to the underlying data through a data-use agreement. Clinical data from the AS Biomarker Cohort is available upon request from the corresponding author (brian.r.lindman@vumc.org). Data from the single-center cardiac magnetic resonance (CMR) AS cohort is available upon request to Marc Dweck (Marc.Dweck@ed.ac.uk). We will aim to respond to requests within 4 weeks of receiving the request. Data from the UK Biobank is available at https://www.ukbiobank.ac.uk. All raw and processed single-nuclear RNA-sequencing data has been deposited on NCBI GEO (GSE262690 [https://www.ncbi.nlm.nih.gov/geo/query/acc.cgi?acc=GSE262690]). Source data are provided with this paper.

## Code availability
All code used for processing single-nuclear data and downstream analyses is deposited at https://github.com/learning-MD/Aortic_stenosis.

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

## Acknowledgements

The authors would like to thank the study participants across all cohorts analyzed. This study was funded by the NIH (R01HL151838; PI: Dr. Elmariah). The contents of this manuscript are solely the authors' responsibility and do not necessarily represent the official views of the granting agencies.

## Author contributions

Conceptualization: B.R.L., S.E., R.S., N.R.T., and E.R.G.; Analysis: A.S.P., M.L.L., K.A., N.K., Q.S., P.L., R.D.P., and E.F.E.; Manuscript writing and revision: B.R.L., A.S.P., K.A., R.V., S.E., E.R.G., W.F.F., S.K., D.J.K., L.G., R.R.M., D.K.G., F.J.M., A.V., N.J., Y.R.S., K.T., T.A., J.E.E., M.N., S.D., Q.S.W., M.R.D., and R.E.G.

## Competing interests

Dr. Lindman is supported by R01HL164526 and R01AG073633 from the NIH, has received investigator-initiated research grant funding and consulted for Edwards Lifesciences, and consulted for Astra Zeneca, Medtronic, Kardigan, and Anteris. Dr. Perry has patents pending for proteomic signatures of fitness, lung, and liver disease. Dr. Amancherla is supported by an American Heart Association (AHA) Career Development Award (#929347), the NIH (K23HL166960), the Red Gates Foundation, and an International Society for Heart and Lung Transplantation Enduring Hearts Transplant Longevity Award. Dr. Amancherla has an institutional disclosure filed for spatial RNA biomarkers of transplant rejection and allograft health. Dr. Gerszten is supported by a Leducq foundation grant (21CVD01). Dr. Nayor supported by R01HL156975 and R01HL131029 from the NIH. Dr. Gillam is an advisor Medtronic, Philips, and Egnite and oversees core lab contracts with Edwards Lifesciences, Medtronic, and Abbott (no direct compensation). Dr Dweck is supported by the British Heart Foundation (FS/SCRF/21/32010) and is the recipient of the Sir Jules Thorn Award for Biomedical Research 2015 (15/JTA). Dr. Das is a founder of Thryv Therapeutics and Switch Therapeutics with equity in both, and has research grants from Bristol Myers Squibb, National Institutes of Health (R35HL 105807). Dr. Gupta is supported by R01HL153607, R01HL154153-03, R01HL148661, R01AG034962, and R01HL145293 from the NIH and is patent holder (#11,079,394) for detection of angiopoietin-2 and thrombospondin-2 for the diagnosis of acute heart failure. Dr. Miller is supported by the Veterans Health Administration, Office of Research and Development, Biomedical Laboratory Research and Development. Dr. Tucker is supported by the NIH R01-HL170051. Dr. Gamazon is a consultant for Thryv Therapeutics, and is a co-inventor on pending patents or disclosures on cardiovascular diseases and phenotypes, and metabolic health, use of RNAs as

therapeutics and diagnostic biomarkers in disease, and methods in metabolomics. Dr. Shah is supported by grants from the National Institutes of Health. Dr. Shah has equity ownership in and is a consultant for Thryv Therapeutics. Dr. Shah is a co-inventor on pending patents or disclosures on molecular biomarkers of fitness, lung disease, cardiovascular diseases and phenotypes, and metabolic health, use of RNAs (including spatial) as therapeutics and diagnostic biomarkers in disease, and methods in metabolomics. Dr. Elmariah is supported by institutional research grants from the NIH (5R01HL151838), the Patient-Centered Outcomes Research Institute, Edwards Lifesciences, Medtronic, and Abbott. Dr. Elmariah is a consultant for Edwards Lifesciences and holds equity and is co-founder of Prospect Health. The remaining authors have no relevant disclosures.

## Additional information

[1]Vanderbilt Translational and Clinical Cardiovascular Research Center, Vanderbilt University School of Medicine, Nashville, TN, USA. [2]Masonic Medical Research Institute, Utica, NY, USA. [3]Department of Biostatistics, Vanderbilt University Medical Center, Nashville, TN, USA. [4]Department of Medicine, Division of Cardiology, Stanford Medical Center, Palo Alto, CA, USA. [5]Department of Medicine, Division of Cardiology, Cleveland Clinic Foundation, Cleveland, OH, USA. [6]Department of Medicine, Division of Cardiology, University of Texas Southwestern Medical Center, Dallas, TX, USA. [7]Department of Cardiovascular Medicine, Morristown Medical Center, Morristown, NJ, USA. [8]Veterans Affairs Tennessee Valley Healthcare System, Nashville, TN, USA. [9]Department of Medicine, Vanderbilt University Medical Center, Nashville, TN, USA. [10]Department of Cardiac Surgery, Vanderbilt University Medical Center, Nashville, TN, USA. [11]Department of Cardiovascular Medicine, Boston University, Boston, MA, USA. [12]Cardiology Division, Massachusetts General Hospital, Harvard Medical School, Boston, MA, USA. [13]BHF Centre for Cardiovascular Science, University of Edinburgh, Edinburgh, UK. [14]Cardiovascular Institute, Beth Israel Deaconess Medical Center, Harvard Medical School, Boston, MA, USA. [15]Broad Institute of Harvard and MIT, Cambridge, MA, USA. [16]Department of Medicine, Division of Cardiology, University of California, San Francisco, CA, USA. [17]These authors contributed equally: Brian R. Lindman, Andrew S. Perry, Michelle L. Lance, Kaushik Amancherla, Namju Kim. [18]These authors jointly supervised this work: Eric R. Gamazon, Nathan R. Tucker, Ravi Shah, Sammy Elmariah. ✉e-mail: brian.r.lindman@vumc.org; tuckern@upstate.edu; ravi.shah@vumc.org; Sammy.Elmariah@ucsf.edu

