## [Transparent Peer Review file · Nature Communications]

Molecular architecture of pressure overload in the human heart: implications on heart failure

Corresponding Author: Dr Ravi Shah

Version 0:

Reviewer comments:

Reviewer #1

(Remarks to the Author)

This study holds significant scientific value as it explores pressure overload and downstream heart failure. The extensive multi-omics analysis provides a valuable resource for this field. The manuscript is logically structured, well-supported by data, and demonstrates a certain level of innovation. However, there are still areas that require improvement to enhance the completeness and persuasiveness of the paper.

1. What are the design principles of the derivation and validation cohorts in the AS Biomarker Study? Was the cohort division based on 825 patients performed randomly?
2. In the proteomics study, were the blood samples collected from peripheral blood? A more detailed description of the blood sample sources is needed.
3. Given that this study aims to map circulating proteomics through the analysis of in situ cardiac tissue, why was bulk-level proteomics of cardiac tissue not performed? Even though snRNA-seq provides higher throughput, tissue-based proteomics remains essential. Particularly considering the study's role as a resource for this field, incorporating tissue-based proteomic data would be highly valuable.
4. The method of obtaining normal control tissues in the snRNA-seq study, especially in comparison to the 11 diseased heart tissue samples, requires a more detailed description.

Overall, this study is meaningful and in-depth. Given the sample size and multi-omics approach, excessive criticism of its "descriptive" nature would be unwarranted. This is an excellent study.

Reviewer #2

(Remarks to the Author)

It is an article showcasing significant work, employing various tools to identify biomarkers. Nonetheless, certain aspects need clarification, particularly regarding the proteomic section. In general, is quite unclear regarding the number of samples used. In proteomic analysis, OLINK technology is employed without specifying the number of samples, whether replicas or pooling are performed, etc. Even though it is theoretically standardized, this information should be explicitly stated, just as it is done in the transcriptomic part, where the "n" is indicated. It would be interesting, considering all the work carried out, to validate some of the proteins identified as potential biomarkers through orthogonal techniques and in an independent cohort of patients than those used in the proteomic analysis.

Reviewer #3

(Remarks to the Author)

This fascinating and potentially highly relevant study uses a comprehensive methodology involving multiple different

modalities, patient groups (and geographies) and techniques to attempt to map out the common principal genetic and multi-omic components that may be common to heart failure syndromes. Conceptually if a specific -omic "signature" could be obtained and easily identified, individuals could potentially be highlighted who are at risk of (but not yet identified with) aortic stenosis and/or heart failure. On this basis, the paper is of merit.

Also, the manuscript is well written with clearly articulated methods, each of which was different from each other and needed adequate explanation (which was given).

I have several comments that may assist the authors:

1. The abstract is complex and confusing. On reading the main manuscript it was clear what the authors were trying to convey, but the abstract on its own was too complex. I suggest a careful rewrite focusing on the specific components, perhaps along the lines of Figure 1.
2. Although the authors were attempting to bring together different diseases and patient groups by using similar techniques to compare (e.g. cross-platform matching Olink to Somalogic via UniProt identifier), showing a common proteome between groups suggests but does not necessarily imply causality. Alternatively, the common association that the authors identified could identify that the presence of each disease (e.g. heart failure or aortic stenosis, and/or other diseases that the authors did not investigate) expresses this signature. In other words, that the presence of this -omic "signature" may not be disease-specific and therefore not clinically helpful if a positive result was found. This possibility should be discussed by the authors.
3. Along similar lines, the demonstration of the unique genetic traits from the HERMES consortium identifies a common trait but this reviewer is not yet convinced they imply causality. A more nuanced discussion may need to be considered. Figures 5F-G, from the evidence presented, do not present a causal link, but they do suggest an association.
4. The results around a potentially simultaneous and aortic stenosis and LV fibrosis disease progression is of great topical relevance. However, the proteomics analysis did not appear to help untangle the uncertainty surrounding this. The results (lines 407-421) would benefit from some more clarity. For instance, although the authors commented on the MRI appearances of myocardial extracellular volume as being indicative of fibrosis, they did not mention extracellular water or deposition as other potential components of the increased extracellular volume component. This would be worth exploring further.
5. In Figure 1, PC1, PC2 and PC3 are not defined. More clarity would be beneficial, particularly since each represented different patterns: Dilated LV morphology (PC1), diastolic dysfunction(PC3), and systolic dysfunction(PC2). Also in Figure 1, the Forrest plot needs more defining, since to this reviewer it appeared that there was a substantially lower risk of mortality in systolic dysfunction(PC2). It is possible PC2 refers to LVEF, and in that case a lower LVEF would increase the hazard. Can the authors clarify this please?
6. Part of the challenge in the interpretation of the data is that heart failure is common, including in patients with AS (e.g. low-gradient AS, where AS may occur "with" heart failure rather than the heart failure being "caused by" AS). Further, the demonstration of additional conditions such as aTTR amyloid, hypertension, and cardiovascular risk factors (along with coronary disease, CKD etc) at the same time as AS and/or heart failure, makes a causal inference from the results obtained even more difficult. It would be helpful if the authors could describe in more detail how these conditions were considered and/or excluded during the analysis.
7. As the authors point out in the discussion, there is also a possibility that treatments could have influenced their findings (e.g. SGLT-2i and/or GLP-1ra), and particularly the possibility that treatments for heart failure may have upregulated some of the proteomics that were examined in the study. This author suggests a little more on this possibility is included in the limitations.

Version 1:

Reviewer comments:

Reviewer #1

(Remarks to the Author)

The authors have addressed or answered the questions I raised very well, and I have no further questions.

Reviewer #2

(Remarks to the Author)

The research presents noteworthy results that contribute valuable insights to the field. The findings are significant and have the potential to impact related areas of study. The work aligns well with established literature, and where applicable, it builds upon previous research effectively. The conclusions and claims are well-supported by the data, with sufficient evidence reinforcing the study's assertions.

Reviewer #3

(Remarks to the Author)

Thank you for the comprehensive responses to my comments and suggestions for modification of the manuscript. I have no further comments or questions.

Reviewer #1 (Remarks to the Author):

1. *Overall, this study is meaningful and in-depth. Given the sample size and multi-omics approach, excessive criticism of its "descriptive" nature would be unwarranted. This is an excellent study.*

Response: We want to thank the Reviewer immensely for these positive comments. As the Reviewer notes, the study was a trans-national effort with extensive phenotyping, multi-omics, and new snRNA-seq and genomics studies. We are heartened that the Reviewers, as a whole, found the study important.

2. *What are the design principles of the derivation and validation cohorts in the AS Biomarker Study? Was the cohort division based on 825 patients performed randomly?*

Response: We appreciate the opportunity to clarify. The AS Biomarker Study was used for 2 main purposes:

- (1) identify proteomic signatures of cardiac remodeling;
- (2) test the associations of those signatures with clinical events.

Our approach to developing proteomics signatures of cardiac remodeling began with using principal component analysis (PCA) to reduce the 12 echocardiographic variables into 3 axes of remodeling (referred to in the manuscript as PC1 - Volumes, PC2 - Systolic Function, and PC3 – Diastolic Function). PCA requires a “complete cases” dataset (meaning all values for echocardiographic parameters for all included subjects had to be quantified, without missingness). In the AS Biomarker Cohort, this condition was present in 519/825 participants. Therefore, we “split” our AS Biomarker Study dataset into the 519 complete cases (no missingness across 12 echocardiographic variables), which we refer to as the “derivation” subset. The remaining 306 cases had missing echocardiographic data for at least one of the measures; we refer to this set as the “validation” subset. We do recognize potential for bias (confounding by indication) in this splitting procedure. To address this potential bias, we compared these two groups (derivation and validation) in **Table 1**. While we observed some statistically significant differences, absolute differences were modest (maximum standardized mean difference 0.227).

3. *In the proteomics study, were the blood samples collected from peripheral blood? A more detailed description of the blood sample sources is needed.*

Response: These were peripheral venous blood samples. We have now edited the **Methods > Proteomics** section to read as follows:

“Peripheral, venous blood samples were obtained prior to TAVI from participants in the AS Biomarker Study, and quantification of the circulating proteome was performed using the Olink Explore 1536 platform in 3 batches.”

4. *Given that this study aims to map circulating proteomics through the analysis of in situ cardiac tissue, why was bulk-level proteomics of cardiac tissue not performed? Even though snRNA-seq provides higher throughput, tissue-based proteomics remains essential. Particularly considering the study’s role as a resource for this field, incorporating tissue-based proteomic data would be highly valuable.*

Response: We agree with the Reviewer completely. As the Reviewer notes, there are some important strengths of snRNA-seq that are unique, including but not limited to whole-transcriptomic interrogation and cell-specific expression profiles. We did not conduct tissue-based proteomics (at a bulk or single cell level [an emerging technology]) *a priori* here, and as such, we do not have that data available. However, this is an area of active interest in our group. In addition, available resources that have measured the proteome in tissues (e.g., GTEx, Human Proteome Atlas) do so in tissues that do not necessarily harbor the disease (in this case, AS). Given the importance of disease induction of the proteome in these settings, mapping to these extant, publicly available resources may not be revealing. Nevertheless, given our agreement with the Reviewer's comments, we have incorporated a statement in our limitations section that attests to this:

"Additionally, proteomic studies at the bulk or single cell level in human myocardium to compare individuals with and without AS would be further revealing beyond the proteo-transcriptional approach presented here."

5. *The method of obtaining normal control tissues in the snRNA-seq study, especially in comparison to the 11 diseased heart tissue samples, requires a more detailed description.*

Response: We have now expanded upon the methods section for tissue procurement as follows:

"Following informed consent, myocardial tissue was obtained with a scalpel from the basal septum of the left ventricle from patients (N=11) undergoing surgical aortic valve replacement (SAVR) for AS or unmatched donor hearts from 2011-2012 (N=9; hearts not used for transplantation) and immediately snap frozen in liquid nitrogen. The interventricular septum was directly visualized by the operating surgeon (e.g., SAVR) or heart procurement surgeon (e.g., donor hearts) prior to tissue procurement. The tissue was subsequently stored in a -80°C freezer until used for analysis. All samples were partial septum samples and non-transmural."

Reviewer #2 (Remarks to the Author):

1. *It is an article showcasing significant work, employing various tools to identify biomarkers. Nonetheless, certain aspects need clarification, particularly regarding the proteomic section.*

RESPONSE: We appreciate the positive comment! We are delighted to have the opportunity to respond.

2. *In general, is quite unclear regarding the number of samples used. In proteomic analysis, OLINK technology is employed without specifying the number of samples, whether replicas or pooling are performed, etc. Even though it is theoretically standardized, this information should be explicitly stated, just as it is done in the transcriptomic part, where the "n" is indicated.*

RESPONSE: We apologize for the lack of clarity. We modified the **Methods > Proteomics** section to include this information:

“Peripheral, venous blood samples (N=825) were obtained prior to TAVI from participants in the AS Biomarker Study, and quantification of the circulating proteome was performed using the Olink Explore 1536 platform in 3 batches. We included 1 sample for each participant. For quality control purposes, 2 pooled plasma samples (pooled from all 825 study participants) were included across all batches (“bridging” samples). A batch effect was detected and corrected by median normalization using the pooled plasma “bridging” samples and setting the batch with the most samples as the reference batch as described.... Overall, 979 circulating proteins were included for analysis in the AS Biomarker Study. Protein levels were expressed in Normalized Protein eXpression (NPX) units, and scaled to mean of 0, and variance 1 across the dataset.”

3. *It would be interesting, considering all the work carried out, to validate some of the proteins identified as potential biomarkers through orthogonal techniques and in an independent cohort of patients than those used in the proteomic analysis.*

RESPONSE: We agree that additional proteomic validation always lends additional support. We interpret “orthogonal techniques and independent cohort” as potential additional populations, methods (e.g., tissue proteomics), or genomics. In this work, we have looked across several populations and platforms and have used some genetic and single cell approaches to bolster several targets. A key translation of our findings is generalization of the AS-related signatures from our AS cohort to a broader population in the UK Biobank. We do not currently have access to similar AS-based cohorts at this size and follow-up. We have included a statement in our Limitations about additional methods that we envision (in accordance with the Reviewer’s important comment) would be relevant:

“Additionally, proteomic studies at the bulk or single cell level in human myocardium to compare individuals with and without AS would be further revealing beyond the proteo-transcriptional approach presented here. Given potential horizontal pleiotropic and confounding effects, definitive conclusions on causality beyond TWAS approaches here will require additional supportive work beyond our TWAS approach (e.g., functional studies). We envision that these approaches can be used to select genes relevant for gain- and loss-of-function approaches in model studies or in vitro organoids for further characterization.”

Reviewer #3 (Remarks to the Author):

1. *This fascinating and potentially highly relevant study uses a comprehensive methodology involving multiple different modalities, patient groups (and geographies) and techniques to attempt to map out the common principal genetic and multi-omic components that may be common to heart failure syndromes. Conceptually if a specific -omic “signature” could be obtained and easily identified, individuals could potentially be highlighted who are at risk of (but not yet identified with) aortic stenosis and/or heart failure. On this basis, the paper is of merit. Also, the manuscript is well written with clearly articulated methods, each of which was different from each other and needed adequate explanation (which was given).*

RESPONSE: We are humbled by these positive comments.

2. The abstract is complex and confusing. On reading the main manuscript it was clear what the authors were trying to convey, but the abstract on its own was too complex. I suggest a careful rewrite focusing on the specific components, perhaps along the lines of Figure 1.

RESPONSE: We appreciate this comment. We have since tried to improve the abstract readability with high fidelity to the conducted studies (now 240 words):

“Pressure overload initiates a series of alterations in the human heart that predate macroscopic organ-level remodeling and downstream heart failure (HF). Here, we study aortic stenosis (AS; pressure overload) through integrated proteomic, tissue transcriptomic, and genetic methods to prioritize targets causal in human HF. First, we identified the circulating proteome of cardiac remodeling in AS, defined by 3 principal components (across 12 echocardiographic traits). This “remodeling proteome” signature specified both known and novel mediators of fibrosis, hypertrophy, and oxidative stress, several of which were associated with cardiac MRI interstitial fibrosis in a separate cohort (N=145). These signatures were strongly related to clinical outcomes in AS (N=802) and in broader populations at-risk for HF in the UK Biobank (N=36,668). We next mapped this remodeling proteome to myocardial transcription in patients with and without AS through single nuclear transcriptomics (snRNA-seq; 20 human hearts, 11 with AS at aortic valve replacement and 9 donor hearts not used for heart transplantation). We observed broad differential expression of genes encoding the remodeling proteome between AS and donor hearts, featuring fibrosis pathways (WNT9A, ITGA6, AGRN, CRIM1, SEMA4C, LAYN, PTX3, HMOX1) and metabolic-inflammatory signaling (ENPP2/ATX, TNF), among others. Finally, through genetic studies leveraging left ventricular myocardial transcription and the circulating proteome, we implicated several targets identified by our prior tissue- and circulating-based approaches as causal in HF. Integrated multi-omic approaches that prioritize both circulating and tissue-level molecular genetic approaches identify targets with causal relevance to HF pathogenesis.”

3. Although the authors were attempting to bring together different diseases and patient groups by using similar techniques to compare (e.g. cross-platform matching Olink to Somalogic via UniProt identifier), showing a common proteome between groups suggests but does not necessarily imply causality. Alternatively, the common association that the authors identified could identify that the presence of each disease (e.g. heart

failure or aortic stenosis, and/or other diseases that the authors did not investigate) expresses this signature. In other words, that the presence of this -omic “signature” may not be diseases-specific and therefore not clinically helpful if a positive result was found. This possibility should be discussed by the authors.

RESPONSE: This comment is an important one. We specifically refrain from causal language in our proteomic association studies, given the issues around confounding that the Reviewer points out (“not disease-specific”). The genetic studies were included to begin to get at causal inference (see response to #4 below). However, we believe that confounding that does not actually invalidate the utility (“not clinically helpful”). Some clinical examples for widely used biomarkers from our cardiovascular clinics for this include:

- C-reactive protein: used occasionally in LDL management as a secondary risk factor to guide toward statin therapy; studied in the JUPITER trial
- B-type natriuretic peptide: can be confounded by age, renal dysfunction, and BMI, among other confounders; however, these levels and their serial trajectory are used in clinical management

In addition, in our survival models in UK Biobank, we adjusted for multiple confounders, though being fully exhaustive in an epidemiologic study is—as the Reviewer sagely points out—impossible for every study. We remain confident that in a robust relation of the “remodeling scores” to outcome, given this was observed in a very large study (UK Biobank) despite common risk factor adjustment.

In response, we have modified our Limitations section to state:

“While we attempted to adjust for multiple confounders in regression modeling, residual confounding remains possible... Nevertheless, the relation of the signature to long-term HF development supports its relevance to HF.”

4. *Along similar lines, the demonstration of the unique genetic traits from the HERMES consortium identifies a common trait but this reviewer is not yet convinced they imply causality. A more nuanced discussion may need to be considered. Figures 5F-G, from the evidence presented, do not present a causal link, but they do suggest an association.*

RESPONSE: We thank the Reviewer for this comment. TWAS aims to identify genes that are causally linked to a phenotype by utilizing the gene’s genetically determined component to determine an association with the phenotype. TWAS integrates GWAS with gene expression studies using genetic instruments to estimate the effect of gene expression (as exposure) on the phenotype (as outcome). TWAS is therefore mathematically related to Mendelian Randomization, a causal inference method that leverages instrumental variables to infer causality from observational genetic data. We do agree that definitive causal conclusions will require additional downstream studies (e.g., functional or randomized controlled trial) in the presence of potential horizontal pleiotropic and confounding effects. We have added this to the Limitations:

“Given potential horizontal pleiotropic and confounding effects, definitive conclusions on causality will require additional supportive work beyond our TWAS approach (e.g., functional studies).”

5. *The results around a potentially simultaneous and aortic stenosis and LV fibrosis disease progression is of great topical relevance. However, the proteomics analysis did not appear to help untangle the uncertainty surrounding this. The results (lines 407-421) would benefit from some more clarity. For instance, although the authors commented on the MRI appearances of myocardial extracellular volume as being indicative of fibrosis, they did not mention extracellular water or deposition as other potential components of the increased extracellular volume component. This would be worth exploring further.*

RESPONSE: We absolutely agree with this, as there are other potentials for CMR ECV elevation, including (as the Reviewer points out) amyloidosis. Generally, the range of ECV is much higher in amyloid (Cuddy et al. JACC CV Imaging 2020; around ECV ≈ 0.50); our ECV range is much lower in this work (ECV $\approx 0.25-0.3$), and in the range of what has been reported for AS (Dusenbery et al. JACC 2014). Within the limits of comparison across ECV methodology, this is an important observation, we feel. In addition, myocardial biopsy routinely in AS is not clinically performed. In the context of our current data availability, further exploration (including other novel measures, such as intracellular lifetime of water, a marker of cardiomyocyte hypertrophy) is not possible.

We have noted in our Limitations section:

“Differential causes of elevated ECV (e.g., amyloidosis) were not explored with endomyocardial biopsy in our study, though our range of ECV was lower than classically reported in this condition.”

6. *In Figure 1, PC1, PC2 and PC3 are not defined. More clarity would be beneficial, particularly since each represented different patterns: Dilated LV morphology (PC1), diastolic dysfunction(PC3), and systolic dysfunction(PC2). Also in Figure 1, the Forrest plot needs more defining, since to this reviewer it appeared that there was a substantially lower risk of mortality in systolic dysfunction(PC2). It is possible PC2 refers to LVEF, and in that case a lower LVEF would increase the hazard. Can the authors clarify this please?*

RESPONSE: We appreciate your help in improving the figures. We added the descriptions of PC1, PC2, and PC3 to the figure legend. Indeed, PC2 refers to systolic function with a positive correlation with LVEF and the Reviewer’s interpretation is correct (e.g., lower LVEF ~ increased mortality). A more detailed version of this figure is shown in **Figure 4**.

7. Part of the challenge in the interpretation of the data is that heart failure is common, including in patients with AS (e.g. low-gradient AS, where AS may occur “with” heart failure rather than the heart failure being “caused by” AS). Further, the demonstration of additional conditions such as aTTR amyloid, hypertension, and cardiovascular risk factors (along with coronary disease, CKD etc) at the same time as AS and/or heart failure, makes a causal inference from the results obtained even more difficult. It would be helpful if the authors could describe in more detail how these conditions were considered and/or excluded during the analysis.

RESPONSE: The reviewer raises several important points. Indeed, almost all patients with AS undergoing aortic valve replacement (AVR) (in our series and in clinical practice) develop heart failure symptoms prior to AVR regardless of LVEF or gradients. It is debated whether these patients with AS would be considered to have a “reversible form of heart failure” (thus “cured” by AVR) or whether they would be deemed heart failure patients with their symptoms in remission after AVR. While some patients in our study may have had LV dysfunction that pre-dated severe AS (what this reviewer is referring to as AS that may occur “with” heart failure), all patients were deemed by the heart team to have severe AS warranting AVR and the vast majority had severe AS with normal LVEF or would have had an LVEF that dropped after the AS became severe. We do not have access to historical longitudinal data to know all these details (i.e., whether the LVEF dropped before or after AS became severe) nor the ancillary testing done by the heart teams to verify that severe AS was present (e.g., dobutamine stress echo, aortic valve calcium score, etc.). Most of these patients had a preserved EF (more similar to HFpEF patients) and a smaller percentage had reduced EF (more similar to HFrEF patients). Further, as the reviewer points out, these patients commonly had other comorbidities and precipitants of LV remodeling/dysfunction and heart failure (e.g., hypertension, coronary disease, etc.). The reality is that patients with AS more often than not have these accompanying comorbidities and almost always develop heart failure symptoms prior to AVR. Nonetheless, we believe that studying patients with AS provides a more pronounced phenotype resulting from severe and chronic pressure overload (on top of the other precipitants of LV remodeling and heart failure) that allows us to make observations that could apply more broadly to other forms of heart failure. To exclude AS patients with any other comorbidities that might influence LV remodeling/dysfunction or heart failure would have made the findings less broadly generalizable. In our Cox models, we did adjust for common co-morbidities such as diabetes, coronary artery disease, atrial fibrillation, eGFR, and blood pressure.

With respect to transthyretin (TTR) amyloid, we do not know whether concomitant amyloid was present as screening for amyloid was not widely performed when this cohort was recruited (2014-2017). The prevalence of amyloid in this patient population has been reported to be approximately 10-15% (*European Heart Journal*, Volume 41, Issue 29, 1 August 2020, Pages 2759–2767).

Accordingly, we have added the following to the limitations:

“Cardiac amyloidosis co-exists in older adults with severe AS in approximately 10-15% of cases and we are unable to parse out the effect of amyloid deposition.”

8. *As the authors point out in the discussion, there is also a possibility that treatments could have influenced their findings (e.g. SGLT-2i and/or GLP-1ra), and particularly the possibility that treatments for heart failure may have upregulated some of the proteomics that were examined in the study. This author suggests a little more on this possibility is included in the limitations.*

RESPONSE: We would be delighted to do this, especially in the context of research results suggesting the potential utility of SGLT2 inhibition in AS (Raposeiras-Roubin et al. Dapagliflozin in Patients Undergoing Transcatheter Aortic-Valve Implantation, NEJM 2025). We have now added a line in our Limitations:

“Additional effects of concomitant medications directed toward HF management may also impact proteomic patterns; further studies adjusted for known and novel medications in AS are warranted.”